# Do We Need More Urban Green Space to Alleviate PM$_{2.5}$ Pollution? A Case Study in Wuhan, China

Yuanyuan Chen [1,2], Xinli Ke [1,*], Min Min [1], Yue Zhang [1], Yaqiang Dai [1] and Lanping Tang [1,3]

[1] College of Public Administration, Huazhong Agricultural University, Wuhan 430070, China; chenyuan2022@webmail.hzau.edu.cn (Y.C.); mmin2010@mail.hzau.edu.cn (M.M.); yuez@webmail.hzau.edu.cn (Y.Z.); yaqiang_dai@webmail.hzau.edu.cn (Y.D.); l.p.tang@webmail.hzau.edu.cn (L.T.)

[2] Department of Sustainable Landscape Development, Institute for Geosciences and Geography, Martin Luther University Halle-Wittenberg, 06120 Halle (Saale), Germany

[3] Department of Environmental Geography, Institute for Environmental Studies, Vrije Universiteit Amsterdam, De Boelelaan 1105, 1081 HV Amsterdam, The Netherlands

* Correspondence: kexl@mail.hzau.edu.cn

**Abstract:** Urban green space can help to reduce PM$_{2.5}$ concentration by absorption and deposition processes. However, few studies have focused on the historical influence of green space on PM$_{2.5}$ at a fine grid scale. Taking the central city of Wuhan as an example, this study has analyzed the spatiotemporal trend and the relationship between green space and PM$_{2.5}$ in the last two decades. The results have shown that: (1) PM$_{2.5}$ concentration reached a maximum value (139 μg/m$^3$) in 2010 and decreased thereafter. Moran's I index values of PM$_{2.5}$ were in a downward trend, which indicates a sparser distribution; (2) from 2000 to 2019, the total area of green space decreased by 25.83%. The reduction in larger patches, increment in land cover diversity, and less connectivity led to fragmented spatial patterns of green space; and (3) the regression results showed that large patches of green space significantly correlated with PM$_{2.5}$ concentration. The land use/cover diversity negatively correlated with the PM$_{2.5}$ concentration in the ordinary linear regression. In conclusion, preserving large native natural habitats can be a supplemental measure to enlarge the air purification function of the green space. For cities in the process of PM$_{2.5}$ reduction, enhancing the landscape patterns of green space provides a win-win solution to handle air pollution and raise human well-being.

**Keywords:** urban green space; particulate matter; spatiotemporal evolution; landscape index; spatial panel regression

## 1. Introduction

Urban green space can include street trees, private gardens, parks, and peri-urban agricultural land within the city [1–3]. Ecosystem benefits provided by green space significantly contribute to maintaining a livable environment for citizens. For instance, air purification [4], alleviation of urban heat islands [5,6], carbon sequestration [7], recreational service [8], and biodiversity conservation [9] can enhance the local quality of life. Small particulate matter with widths less than 2.5 μm (PM$_{2.5}$) is one of the outcomes of anthropogenic activities [10,11]. PM$_{2.5}$ damages human health in terms of the cardiovascular system [12], respiration [13,14], and mortality [15]. In contrast, green spaces directly absorb particulate matter through the retention ability of leaves [16] and slow down the spread of PM$_{2.5}$. However, green space is often affected by urban development, which exacerbates the correlations with PM$_{2.5}$ in areas where population density is already high.

The adverse effects of PM$_{2.5}$ concentrations are a global concern. According to the guidelines released by the WHO, annual exposure to PM$_{2.5}$ concentrations under 5 μg/m$^3$ has a less negative impact on humans [17]. The guidelines and actual emissions of PM$_{2.5}$ in many countries and regions have not yet adopted or attained the same standard yet.

For example, the two levels of PM$_{2.5}$ concentration identified in China are 15 μg/m$^3$ and 35 μg/m$^3$ (annually) [18]. According to the National Ambient Air Quality Standard (NAAQS), the PM$_{2.5}$ concentration in India should be no more than 40 μg/m$^3$ (annually) [19]. The limit of PM$_{2.5}$ set by the European Commission is 20 μg/m$^3$ (annually) [20]. There are gaps between the different guidelines (Appendix A, Table A1), which indicates that many efforts need to be made in global PM$_{2.5}$ reduction. Notably, the most dramatic influence of air pollution is the effect on longevity. Ebenstein et al. (2017) claimed that a 10.35 μg/m$^3$ increase in PM$_{10}$ reduced life expectancy by 7.68 months [21]. If PM$_{2.5}$ concentrations reach WHO air quality guidelines (annual 10 μg/m$^3$, released in 2005) in all countries, then life expectancy will increase by a population-weighted median of 7.20 months [22]. Moreover, poor air quality exposure increases the rates of cardiovascular diseases [23,24]. In addition, regional air quality can affect citizens' self-reported happiness [25].

Governments and research institutes construct PM$_{2.5}$ datasets to examine the concentration, possible origins, and the main influencing factors. Ground air quality stations observe hourly, daily, monthly, and yearly results of particulate matter measurements. These datasets are widely applied in spatial and temporal change analysis of PM$_{2.5}$ [26–28] and are combined with Aerosol Optical Depths (AODs) data to generate full cover maps across regional and global scales [29,30]. PM$_{2.5}$ concentrations and distributions are affected by anthropogenic and natural factors. The emissions from burning biomass and fossil fuels are the main origins of particulate and gaseous pollutants [31]. Landscape fires are a significant PM$_{2.5}$ source in many regions [32]. Meteorological factors significantly influence the distribution of PM$_{2.5}$ through wind speed and temperature [33]. Urbanization, road density, and urban morphologies correlate with PM$_{2.5}$ pollution [34,35]. Land use and cover change also contribute to the distribution of PM$_{2.5}$ [36]. Methods such as land-use regression, random forest, and the spatial panel model have been widely applied in examining the influencing factors and predicting the features of PM$_{2.5}$ concentration [37–40].

Green space is one of such factors that can influence regional PM$_{2.5}$ concentration, decreasing the concentration by deposition from leaves, affecting wind speed, and impacting air humidity [41,42]. Morphological and physiological traits of green space also make a difference in PM$_{2.5}$ reduction [43]. Higher green space core and bridge proportions significantly reduce the PM$_{2.5}$ concentration [44]. The leaf area index negatively correlates with PM$_{2.5}$ mass concentration [45]. The air purification effects of green space have a particular functional scale. Considering the circular buffers of 0.5–5 km in radius, centered in the monitoring stations, the total edge length of green space has a greater impact on PM$_{2.5}$ concentration than the green cover area. However, the influence of green cover dominates on the 3–5 km scale [46]. In addition, biogenic volatile organic compounds (BOVC) emissions of green space-induced PM$_{2.5}$ concentration are harmful to human health [47]. The dual role of green space in warm seasons is to both remove air pollution and enhance air quality [48], which needs to be pointed out in the correlation analysis of PM$_{2.5}$ concentration and green space morphology.

During the process of urbanization, urban green space is inevitably influenced by building encroachment. Decreasing its area will further undermine the ecosystem functions [49], especially for the central city area where impervious land occupies a higher proportion. Air purification, recreation, and habitat maintenance services of green spaces play significant roles in residents' well-being. Thus, the correlation of green space landscape patterns on PM$_{2.5}$ concentration across an extended period needs to be clarified. Because PM$_{2.5}$ can migrate within the regional area within a particular time, long time-series data of PM$_{2.5}$ concentrations are more useful to develop annual full cover maps [50]. Furthermore, these maps can be fitted to examine the relationship between PM$_{2.5}$ and green space landscape patterns on a grid scale.

This study focuses on the central city of Wuhan because it has undergone rapid urbanization and drastic changes in air quality during the last 20 years. The main objectives of the present work are to (1) reveal the spatial-temporal changes in green space and PM$_{2.5}$ concentration and to (2) identify the correlations between landscape features of green space

and $PM_{2.5}$ concentration. Finally, we propose strategies to enhance the positive relationship between $PM_{2.5}$ and green space.

## 2. Data and Methodology

### 2.1. Study Area

Wuhan is the capital of Hubei province, the geographical and economic center of the middle part of China, and it has a total area of 8569.15 km$^2$ (Figure 1). It is located in a humid subtropical climate zone, and the Yangtze River flows across the central city. The annual precipitation is 2012.4 mm, and the average temperature is about 17.2 °C (2020). The total population is about 12.44 million [51], which increased by 54.67% from 2000 to 2020. Wuhan has undergone rapid urbanization over the past two decades, especially in its central districts where 57.88% of the population (7.21 million) lives.

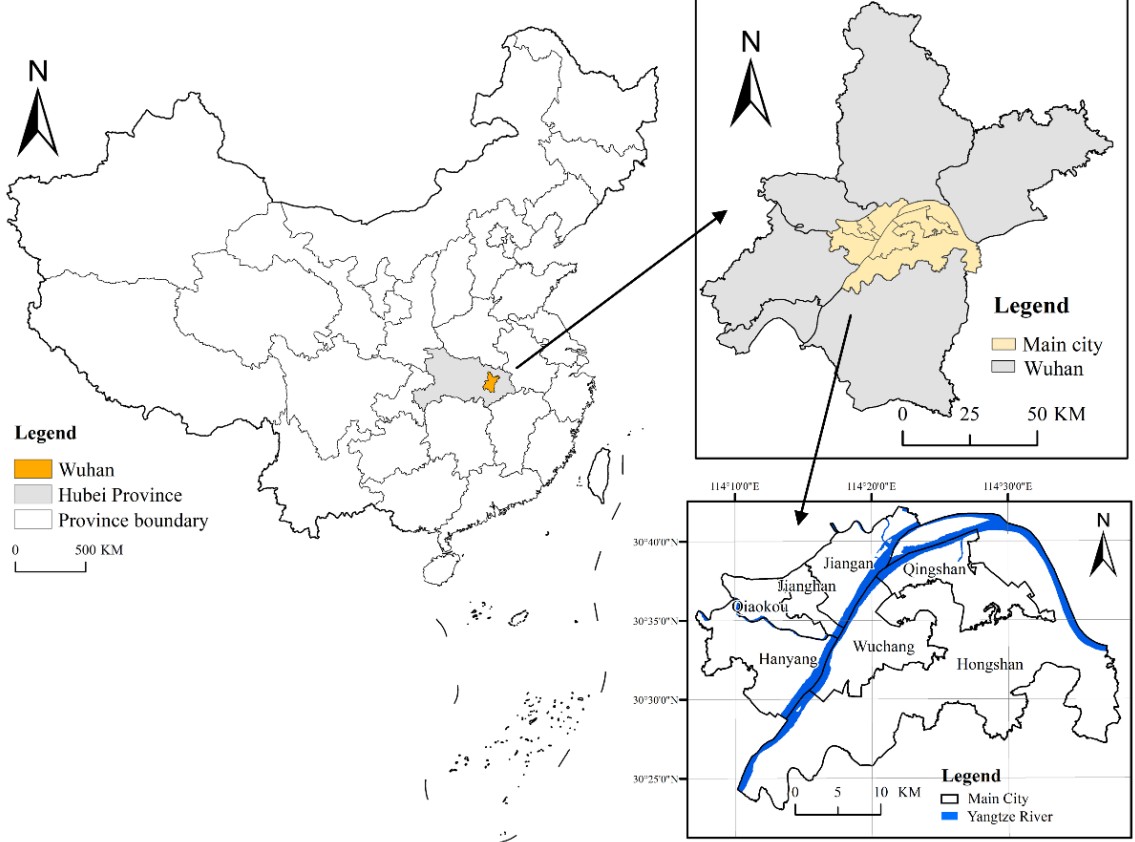

**Figure 1.** Location of the study area.

Expanding the impervious surface threatens the distribution pattern and amount of green space. Factory production, fossil fuel consumption, and private cars are all growing along with urban development, which causes urban problems such as air pollution and traffic jams. More attention has been given to these negative outputs and the importance of protecting the natural environment in the central districts. The local government has implemented a series of measures to deal with these problems, such as reducing energy consumption and unqualified fuels of private cars. The Chinese government gives priority status to the promotion of air quality. A $PM_{2.5}$ concentration reduction by 10% is promised in China's 14th Five-Year Plan (2021–2025) [52], and rules such as restricting high-polluting sources and constant air quality monitoring will continue.

### 2.2. Data for Analysis

This study represented green space as areas composed of land cover, including woodland, grassland, and cropland. There was a rapid increase in construction land use in

the central districts, so we also considered artificial surfaces. According to the current land use classification in China (GB/T 21010-2017) [53], the land cover types were manually interpreted from Google history remote sensing images with a resolution of 2 m (2000/2010/2020) [54]. To improve the precision of our interpretation, we took the land cover maps generated from GlobalLand30 as a reference [55]. The total accuracy of GlobalLand30 is 85.72%, and the Kappa coefficient is 0.82. We calculated landscape patterns of green space in 2020. For coherence with the following data, we described the results of green space landscape patterns as in 2019.

PM$_{2.5}$ concentration data (1 km resolution, annual average) were obtained from an open-access database [56], which includes long-term, full-coverage, and high-resolution ground-level air pollutants in China. The local government of Wuhan implemented lockdown strategies from 23 January–22 February 2020. Consequently, this study applied the PM$_{2.5}$ dataset of 2019 in the spatial panel regression to avoid the lockdown effect. Thus, this study extracted annual PM$_{2.5}$ concentration data from 2000, 2010, and 2019.

The spatial maps of gross domestic product (GDP) were drawn from the Data Center of Resources and Environmental Sciences (RESDC) [57]. The resolution of GDP maps in 2000, 2010, and 2015 was 1 km. Because the GDP for 2019 was unavailable, it was estimated by multiplying 2015 data by the growth rate from 2015 to 2019.

### 2.3. Methodology

2.3.1. Landscape Patterns of Green Space

The configuration and composition of green space can influence the deposition rate of particulate matter. Hence, we applied the total area (TA), contagion index (CONTAG), largest patch (LPI), and Shannon's diversity index (SHDI) to demonstrate the spatial distribution of green space (Table 1). TA aims to reflect the area of green space. LPI represents the dominating patches occupying the proportion of green space area. CONTAG, as the fractal dimension index, describes the spatial patterns of patches. Finally, SHDI can manifest the complexity of green space types.

**Table 1.** Formula and explanations of landscape indexes.

| Landscape Indexes | Formula | Description | References |
|---|---|---|---|
| Total area (TA) | $TA = \sum_{j=1}^{n} a_{ij}$ | Equals the sum of green space areas of patches. | [58,59] |
| Largest patch index (LPI) | $LPI = \frac{max(a_{ij})}{A} * 100$ | Equals the percentage of the landscape comprised by the giant patch | [60,61] |
| Contagion index (CONTAG) | $CONTAG = \left(1 + \frac{\sum_i^n \sum_j^n (P_i*m - \ln(P_i*m))}{2\ln(n)}\right) *100$ | Ranging from 0 to 100. The lower the index value, the more scattered the urban landscape pattern and the higher the average degree of fragmentation. | [62–64] |
| Shannon's diversity Index (SHDI) | $SHDI = -\sum_{i=1}^{n} (P_i * lnP_i)$ | This represents diversity. The value increases as the number of different patch types increases. | [65,66] |

Note: $a_{ij}$ is the area (m$^2$) of patch ij; A is the total area of the grid (m$^2$); n is the number of patches; $P_i$ represents the proportion of landscape occupied by patch type i; $g_{ik}$ is the number of adjacencies (joins) between pixels of patch types (classes). $m = g_{ik} / \sum_{k=1}^{n} g_{ik}$.

A fishnet (grids with 1 km resolution) was created from the PM$_{2.5}$ dataset (ArcGIS 10.5, "create fishnet" tool) and applied in the extraction of green space. The landscape indexes were calculated by Fragstats (4.2).

2.3.2. Spatial Correlation Analysis

(1)    Spatial weight matrix

The weight matrix is based on the queen contiguity criterion to reflect the neighboring influence. Grids with the common vertexes and edges are assigned to 1, and otherwise 0. The formula is as follows:

$$W_{ij} = \begin{cases} 1, & i \text{ shares a corner or edge with } j; \\ 0, & i \text{ isn't adjacent to } j. \end{cases} \tag{1}$$

(2)    Moran's I and hotspot analysis

We conducted the spatial autocorrelation test to examine the spatial interdependence of $PM_{2.5}$ concentration. Moran's I was applied to calculate the degree of spatial autocorrelation [67,68]. The formula is expressed as follows:

$$I = \frac{\sum_{i=1}^{N} \sum_{j=1}^{N} W_{ij}(x_i - \overline{x})(x_j - \overline{x})}{s^2 \sum_{i=1}^{N} \sum_{j=1}^{N} W_{ij}} \tag{2}$$

$$S^2 = \frac{\sum_{i=1}^{N}(x_i - \overline{x})^2}{N} \tag{3}$$

where I is the univariate global spatial correlation index; $x_i$, $x_j$ are the attribute values of features $i$ and $j$, respectively; $\overline{x}$ is the average value of features $i$ and $j$; $S^2$ is the sample variance; $W_{ij}$ is the spatial weight between features $i$ and $j$; and $N$ equals the total number of features.

For the Moran's I analysis, the null hypothesis states that the attribute being analyzed is randomly distributed among the features in the study area. The null hypothesis may be rejected when the *p*-value is lower than the defined threshold (5% in this study) and the z-score is positive. The spatial distribution of high values and low values in the dataset is more spatially clustered. When the *p*-value is lower than the defined threshold and the z-score is negative, the null hypothesis may also be rejected. In that case, the spatial distribution of high values and low values in the dataset are more spatially dispersed.

Hotspot analysis is applied to identify the clustering attribute of factors [69,70]. This analysis creates maps of statistically significant hot and cold spots. The hotspot analysis calculates the Getis–Ord $G_i^*$ statistic for each feature in a dataset:

$$G_i^* = \frac{\sum_{j=1}^{N} w_{i,j}x_j - \overline{x}\sum_{j=1}^{N} w_{i,j}}{S\sqrt{\frac{(N\sum_{j=1}^{N} w_{i,j}^2 - (\sum_{j=1}^{N} w_{i,j})^2)}{N-1}}} \tag{4}$$

The explanations of variables are the same as in Formula (2). $G_i^*$ is a z-score that represents the standard deviation. The resultant z-scores and *p*-values show where features with high or low values cluster spatially. For statistically significant positive z-scores (5% in this study), the larger z-scores represent intense clustering of high values (hot spot), and vice versa. The quadrants were divided as "High-high (H-H)", "Low-low (L-L)", "High-low (H-L)", and "Low-high (L-H)".

2.3.3. Spatial Panel Model

Variance inflation factors (VIFs) were calculated to exclude the highly correlated variables from a stepwise procedure. First, we assumed that relevant factors influenced the $PM_{2.5}$ concentration without spatial dependency. A basic Ordinary Least Squares (OLS) model is expressed as follows:

$$Y_{it} = \beta_0 + \beta X_{it} + \varepsilon_{it} \tag{5}$$

where $Y_{it}$ is the PM$_{2.5}$ concentration in cell $i$ at time $t$; $\beta_0$ is a constant term; $\beta$ is the parameter of $X_{it}$; $X_{it}$ represents the independent variables, referring to the landscape indexes of green space, GDP, and artificial surface proportion in this study; and $\varepsilon_{it}$ is the error term.

Ignoring the spillover effect of PM$_{2.5}$ concentration might lead to bias. The spatial lag model (SLM) presumes that dependent variables interact among neighbors [71,72]. Thus, we changed Formula (5) into:

$$Y_{it} = \beta_0 + \rho \sum_{j=1}^{N} W_{ij} Y_{it} + \beta X_{it} + \varepsilon_{it} \tag{6}$$

where $\rho$ is the spatial lag effect coefficient. When the error terms correlate, the random effect affects the spillover effect [73]. The Spatial Error Model (SEM) is expressed as follows:

$$Y_{it} = \beta_0 + \beta X_{it} + \mu_{it} \tag{7}$$

$$\mu_{it} = \lambda W \mu_{it} + \varepsilon_{it} \tag{8}$$

where $\lambda$ represents the average degree of spatial correlations of errors. Furthermore, the Spatial Durbin Model (SDM) measures the effect of neighboring dependent and independent variables [74]. In this case, PM$_{2.5}$ concentration is influenced not only by the PM$_{2.5}$ value of neighbors but also by the neighboring independent variables:

$$Y_{it} = \beta_0 + \rho \sum_{j=1}^{N} W_{ij} Y_{it} + \beta X_{it} + \theta \sum_{j=1}^{N} W_{ij} X_{ijt} + \varepsilon_{it} \tag{9}$$

where $Y_{it}$ is the PM$_{2.5}$ concentration and $\theta$ is the spatial spillover effect coefficient of the independent variables.

The Likelihood Ratio test (LR test) determines whether SLM, SEM, and SDM models are more fitted for the green space and PM$_{2.5}$ data. The Hausman test was conducted to select between the fixed-effects (FE) or random-effects (RE) panel model [75]. Akaike's Information Criteria (AIC) and Bayesian Information Criteria (BIC) supported the model selection [76]. The lower the AIC and BIC values, the better the model fits the data. The spatial panel model was processed in Stata15 (package "xsmle").

## 3. Results

### 3.1. Temporal Change and Spatial Distribution of Green Space from 2000 to 2019

The amount of green space (TA) decreased from 381.04 km$^2$ to 282.60 km$^2$, or a decrease of 25.83%, from 2000 to 2019. Conversely, artificial surfaces increased from 290.54 km$^2$ to 387.68 km$^2$ in the same period, increasing by 33.43%. The adverse direction of transformation reflects the observation that human activities have an intervention effect on the natural environment. Consistent with the downward trend of TA, both LPI and CONTAG decreased, while SHDI increased (Table 2).

**Table 2.** Landscape indexes of green space in the central city from 2000 to 2019.

| Year/Landscape Indexes | TA (km$^2$) | LPI (%) | CONTAG (%) | SHDI |
|:---:|:---:|:---:|:---:|:---:|
| 2000 | 381.04 | 6.05 | 63.28 | 0.78 |
| 2010 | 323.52 | 5.93 | 54.81 | 0.96 |
| 2019 | 282.60 | 3.35 | 53.14 | 1.00 |
| changing rate (%) | | | | |
| 2000–2010 | −15.10 | −2.04 | −13.39 | 23.17 |
| 2010–2019 | −12.65 | −43.40 | −3.04 | 4.37 |
| 2000–2019 | −25.83 | −44.56 | −16.03 | 28.55 |

The changing rates of landscape indexes in the earlier 10 years (from 2000 to 2010) were higher than those in later years (from 2010 to 2019), except for LPI. The evolution of green space resulted in an evident reduction in total area and large patches, which led to fewer connections of natural land cover. There was an apparent downward trend in green space in the urban periphery and an incremental trend in the center area (Figure 2). In addition, landscape indexes demonstrated a clustering pattern (Appendix A, Figure A1). Moran's I for TA was in a downward trend due to the loss of ample green space around the edge of the central city. The spatial correlation of CONTAG kept increasing while LPI and SHDI decreased from 2000 to 2019. This result suggested that green space has become more fragmented and less diverse.

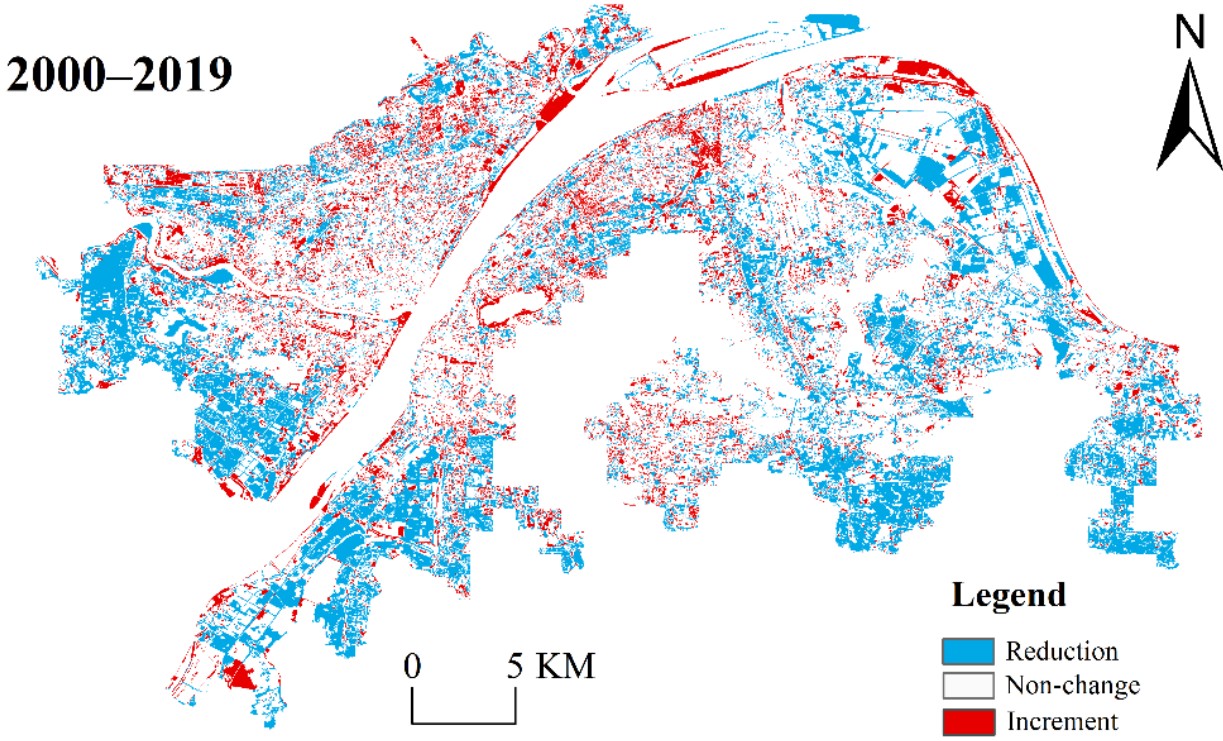

**Figure 2.** Spatial change in green space from 2000 to 2019 in the central city of Wuhan (reduction in blue represents the green space occupied by other types of land use; increment is the area transferred from other land use types to green space).

### 3.2. Spatial and Temporal Changes in $PM_{2.5}$ Concentration

From 2000 to 2019, the $PM_{2.5}$ concentration underwent a transition from an upward to a downward trend. The annual average values of $PM_{2.5}$ for 2000, 2010, and 2020 were 75.20 μg/m$^3$, 85.87 μg/m$^3$, and 46.10 μg/m$^3$, respectively (Figure 3). Compared to 2000, a higher level of $PM_{2.5}$ concentration in 2010 led to an increase in the average $PM_{2.5}$ by 14.19%. The maximum value occurred in 2010. The dense $PM_{2.5}$ concentration clustered around the city center. $PM_{2.5}$ concentration decreased by 46.33% from 2010 to 2019, which corresponds with the promotion of air quality across the region. The distribution of $PM_{2.5}$ was in a clustering pattern according to the global Moran's I value from 2000 to 2019 (see Appendix A, Figure A1).

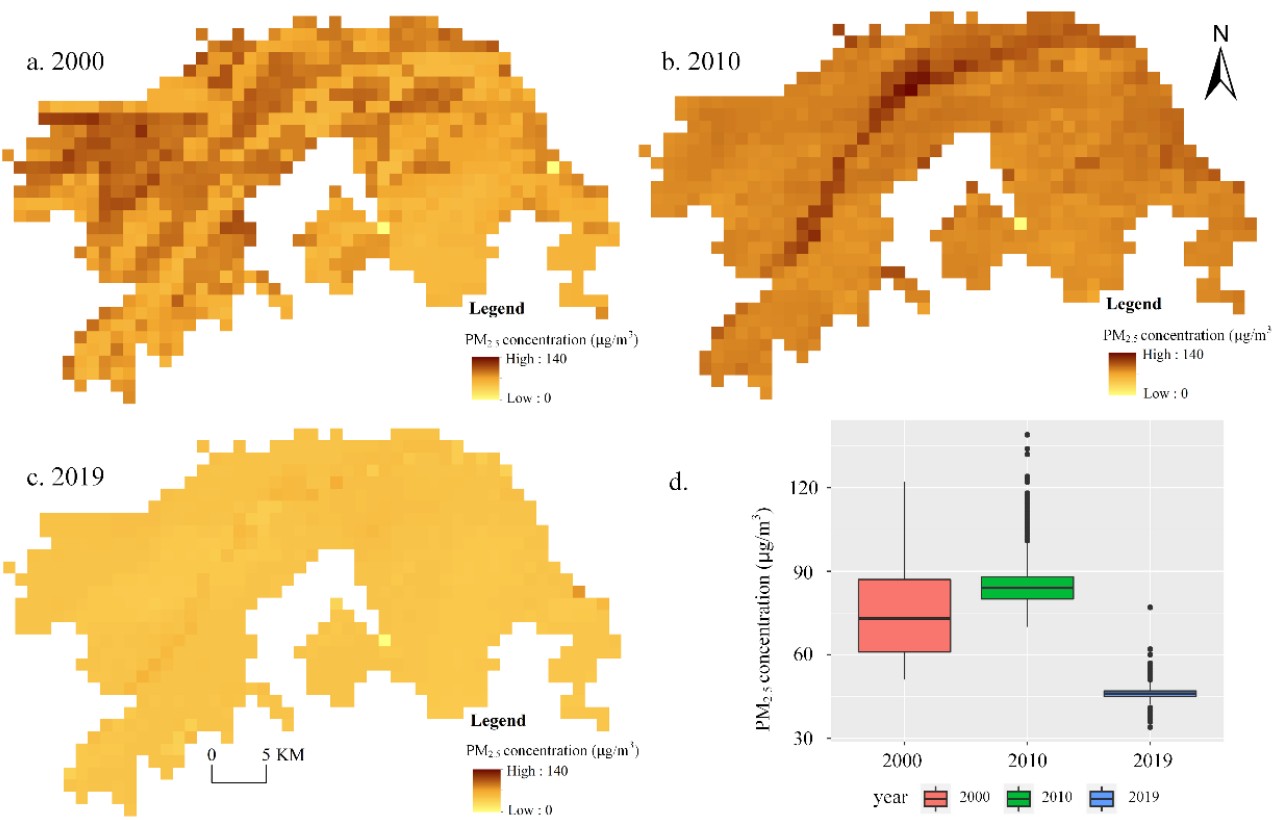

**Figure 3.** Spatial and temporal changes in annual PM$_{2.5}$ concentrations in the central city of Wuhan from 2000 to 2019 (**a**–**c**): PM$_{2.5}$ concentration maps were extracted from the annual PM$_{2.5}$ concentration maps of China [56]; (**d**): boxplot of PM$_{2.5}$ concentration. The interquartile range is from the 25th percentile to the 75th percentile. The 50th percentile is the median, represented by the middle black line).

### 3.3. Spatial Patterns of Green Space and PM$_{2.5}$ Concentration

The local spatial correlations among PM$_{2.5}$ concentrations and landscape indexes are shown in Figure 4. The number of grids with significant "H-H", "H-L", "L-H", and "L-L" are demonstrated in the four quadrants. From 2000 to 2019, the spatial scale of "H-H" and "H-L" cluster patterns decreased, which indicates that both PM$_{2.5}$ and green space were downward. "L-L" group numbers increased on the left-hand side of the Yangtze River, while "L-H" cells clustered on the right-hand side. There were spatial correlations between PM$_{2.5}$ and green space landscape indexes. However, clarifying how the landscape patterns correlated with PM$_{2.5}$ requires further regression analysis.

### 3.4. Correlations of Green Space Landscape Patterns and PM$_{2.5}$ Concentration

We also considered GDP and the artificial surface area because PM$_{2.5}$ is affected by economic development and urbanization [33,34]. The VIF value of independent variables was less than 10. There was a low correlation among variables and the fit for regression analysis.

As the results have shown (Table 3), ordinary linear regression and spatial panel models examined the correlations between landscape indexes and PM$_{2.5}$ from different aspects. The R-square of OLS (0.435) showed that the landscape patterns of green space and GDP could explain the PM$_{2.5}$ concentration to a certain degree. Meanwhile, the artificial surface demonstrated an insignificant correlation with PM$_{2.5}$. GDP and PM$_{2.5}$ concentration significantly correlated. TA and LPI was negatively correlated with PM$_{2.5}$, which indicates that the area of green space and larger patches could contribute to the air purification. CONTAG demonstrated an insignificant relationship with PM$_{2.5}$ concentration. Finally, SHDI showed a strong influence in alleviating PM$_{2.5}$ concentration.

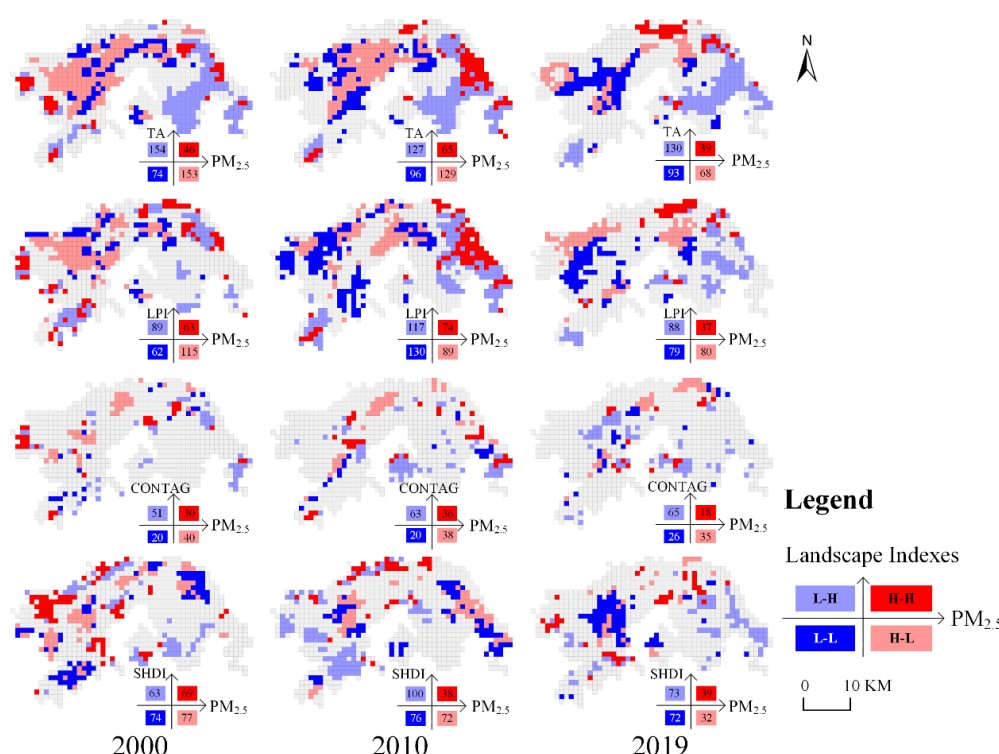

**Figure 4.** Local spatial correlations of PM$_{2.5}$ and landscape patterns in the last two decades. (As mentioned in the methodology, "H-H" represents the high value of landscape indexes and PM$_{2.5}$ concentration clusters in the spatial scale. The number inside of the rectangle is the calculation of grids, and for example, the number of "H-H" of SHDI and PM$_{2.5}$ in 2000 is 69).

Spatial panel models further manifested the spatial correlations of PM$_{2.5}$ and green space. LPI and GDP negatively correlated with PM$_{2.5}$ in SLM and SEM. Neighboring GDP and TA showed a negative correlation with PM$_{2.5}$ concentration in the results of SDM. To determine a model for further analysis, we examined the statistically significant LR and Hausman test, together with lower AIC and BIC values. We found that the spatial lag model (SLM) was more adequate. In detail, LPI negatively correlated with PM$_{2.5}$, which suggests the importance of protecting natural sizeable green space. Even though SHDI showed the most effective reduction in PM$_{2.5}$, in addition to the effect of GDP, the *p*-value was not significant. TA and CONTAG correlations with PM$_{2.5}$ were unclear.

**Table 3.** Regression analysis of PM$_{2.5}$ concentration (dependent variable) and landscape indexes, GDP, artificial surfaces.

| Models<br>Coefficients | OLS | SLM | SEM | SDM |
|---|---|---|---|---|
| Constant | 169.218 | - | - | - |
| Artificial surfaces | 0.000 | −0.009 | 0.002 | 0.006 |
| GDP | −10.204 *** | −1.601 *** | −1.630 *** | −0.486 |
| TA | −0.15 *** | 0.006 | 0.032 ** | 0.030 ** |
| LPI | −0.072 *** | −0.023 *** | −0.027 *** | −0.028 *** |
| CONTAG | −0.002 | −0.007 | −0.010 | −0.008 |
| SHDI | −3.64 *** | −0.597 | −0.510 | −0.638 |

**Table 3.** *Cont.*

| Coefficients / Models | OLS | SLM | SEM | SDM |
|---|---|---|---|---|
| W * Artificial surfaces | - | - | - | −0.022 |
| W * GDP | - | - | - | −1.324 *** |
| W * TA | - | - | - | −0.065 *** |
| W * LPI | - | - | - | 0.026 |
| W * CONTAG | - | - | - | 0.023 |
| W * SHDI | - | - | - | −0.197 |
| $\rho$ | - | 0.876 *** | - | 0.868 *** |
| $\lambda$ | - | - | 0.938 *** | - |
| Residual variance | - | 30.191 *** | 29.901 *** | 30.145 *** |
| R-square | 0.435 | 0.4254 | 0.2900 | 0.4872 |
| AIC | 20,268.721 | 15,859.38 | 15,965.58 | 15,853.29 |
| BIC | 20,309.385 | 15,905.85 | 16,012.05 | 15,934.61 |

Note: ***, **, * represent the significance level of *p*-value 1%, 5%, and 10%, respectively.

## 4. Discussion

This paper has focused on the relationship between green space landscape patterns and $PM_{2.5}$ concentration in Wuhan central city over the last two decades. The significant negative correlation between LPI and $PM_{2.5}$ was examined in the regression analysis. Reserving naturally sizeable green space and enlarging present green space are promising measures to support the mitigation of $PM_{2.5}$. However, impervious land constantly encroaches on surrounding green space [77]. The present study found that almost a quarter of green space vanished in the central city of Wuhan from 2000 to 2019. The loss of marginal green space exceeds the increment in the city center (Figure 2), which reduces the total area [78].

From 2010 to 2019, $PM_{2.5}$ decreased by 46.33% in the central city of Wuhan. Policies such as the Air Pollution Prevention and the Control Action Plan, which was released by the Chinese government in 2013, significantly decreased the population-weighted annual mean $PM_{2.5}$ concentrations in China from 61.8 to 42.0 $\mu g/m^3$ within five years [79]. Anthropogenic emission dominated the reduction. National policies could accelerate the efforts of local authorities to improve the air quality for sustainable development. In the central city of Wuhan, measures such as promoting green space coverage, reducing the use of coal, and green energy consumption have gradually alleviated regional $PM_{2.5}$ pollution [80]. Indeed, the average $PM_{2.5}$ concentration decreased and reached 46.1 $\mu g/m^3$ in 2019. However, this is still far from the guideline (5 $\mu g/m^3$) suggested by the WHO to reduce the harmful impacts on human health and longevity [17].

Although artificial surfaces did not show an evident relationship with $PM_{2.5}$ concentration, the economic activities demonstrated a statistically significant correlation with $PM_{2.5}$. According to the Environmental Kuznets Curve, a post-urbanization period of economic development can exert a less negative influence on the environment [81,82]. In our case, during the primary period of urbanization, economic development dominated society. Factories with polluting emissions near the central city center led to poor air quality. Later, advanced economic development and urban planning would reduce the disturbance to air quality. This interpretation might explain the cluster of the "L-L" group on the left-hand side of the Yangtze River, where high urbanization and a low area of green space occur.

Notably, the relationship between green space and $PM_{2.5}$ needs to be carefully interpreted. First, many anthropogenic and meteorological factors mutually affect $PM_{2.5}$ concentration. The contribution rate of the main urban area source to $PM_{2.5}$ concentration exceeds 60% in Wuhan [83]. Humidity and temperature can also contribute to the level of $PM_{2.5}$ [84]. Even though green space features showed significant correlations with $PM_{2.5}$ in spring [85], this mechanism is complicated. Meanwhile, biogenic volatile organic compound emissions from green space and other meteorological conditions might influence $PM_{2.5}$ mitigation. In the context of regional climate and air quality model simulations,

vegetation emissions led to an increase in $PM_{2.5}$ concentration over 10 μg/m$^3$, which occupied around 20% of the average observed $PM_{2.5}$ concentration [86]. Vegetation emissions can also increase the particle fluxes in urban environments [87]. Thus, the dual effects of vegetation need to be considered in increasing green space to achieve $PM_{2.5}$ mitigation.

Green space landscape patterns are stressed in the present study for two reasons. First, according to previous studies, the green space area and diversification of vegetation could exert great significance in air regulation [88]. SHDI also showed a negative correlation with $PM_{2.5}$ concentration in this study. The amount of $PM_{2.5}$ adsorption by leaves demonstrated a noticeable seasonal difference. Conifers were more effective in $PM_{2.5}$ adsorption than broadleaf trees were [89]. Green Infrastructure (GI) practices could significantly improve air quality in urban areas [90]. Vegetation structure, composition, and management are essential to optimize the capacity of green spaces to purify air quality [91]. The biophysical attributes of green spaces directly contribute to the reduction in $PM_{2.5}$. Second, the ecosystem services from green space combined with $PM_{2.5}$ reduction could mutually affect the health of the residents [92]. Expanding the green area and reducing fragmentation can support the improvement of air quality and decrease the mortality rate of the citizens. Combined with the results of this study, large green space patches deserve more attention in urban land use planning.

This study examined the correlations between green space morphology and $PM_{2.5}$ concentration over a long period. However, there are some limitations. First, during the lockdown period, the $PM_{2.5}$ concentration of Wuhan decreased by 27 μg/m$^3$ compared with the same period in 2019 [93]. In total, 90% of the reduction resulted from the lockdown because $PM_{2.5}$ mainly comes from factories, coal consumption, and transportation. A series of research studies in India [94], America, Italy, France [95], Southern Italy [96], the United Kingdom [97], Barcelona [98], and China [99] have shown positive outcomes of promoting air quality during the same period. Although the lower pollution level briefly appeared during the lockdown, the positive effects will guide decision-makers to make more efforts to moderate $PM_{2.5}$ concentration from the origins [100,101]. Correlations between green space and $PM_{2.5}$ concentrations should be further investigated under different policies.

Although this study showed the possible correlations between green space morphology and $PM_{2.5}$ concentration, the interpretation was limited by the absence of $PM_{2.5}$ sources and meteorology data. Moreover, the physical mechanisms of green space and $PM_{2.5}$ concentration need to be further explained. For instance, deposition on leaves might offset the emissions of biogenic volatile organic compounds, even though the net $PM_{2.5}$ deposition of green space remains. In that case, the contribution of green space to decrease the $PM_{2.5}$ concentration will be more comprehensive.

## 5. Conclusions

Urban green space landscape patterns have shown different correlations with $PM_{2.5}$. In the present study, LPI and SHDI negatively correlate with $PM_{2.5}$ concentrations. Hence, it is rational to implement measures to strengthen the advantageous landscape features of green spaces. However, under the pressure of urbanization, only a limited quantity of land is left for expanding green spaces. Our results show that green space has decreased by a quarter in the Wuhan central city over the last two decades. Therefore, enhancing landscape patterns of green space deserves more attention. In terms of $PM_{2.5}$ concentration, preserving larger natural patches and new increments around the existing green area are promising supplementary measures for locations with a lower proportion of green space and higher pollution emissions. These findings may serve as a reference for cities undergoing land-use conflicts and experiencing air pollution.

**Author Contributions:** Y.C.: data curation, methodology, writing—original draft preparation. X.K.: conceptualization, funding acquisition, supervision. M.M.: investigation, writing—review and editing. Y.Z. and Y.D.: software, visualization. L.T.: manuscript review. All of the authors contributed to improving the quality of the manuscript. X.K. is responsible for the academic opinion of this manuscript. All authors have read and agreed to the published version of the manuscript.

**Funding:** This study was funded by the National Natural Science Foundation of China (grant number 41971240), National Social Science later stage Foundation of China (grant number 19FGLB071), and The Ministry of Education of Humanities and Social Science project (grant number 18JHQ081).

**Institutional Review Board Statement:** Not applicable.

**Informed Consent Statement:** Not applicable.

**Data Availability Statement:** Not applicable.

**Acknowledgments:** The authors sincerely thank the anonymous reviewers for their insightful comments and suggestions. We also gratefully acknowledge the financial support from China Scholarship Council.

**Conflicts of Interest:** The authors declare no conflict of interest.

## Appendix A

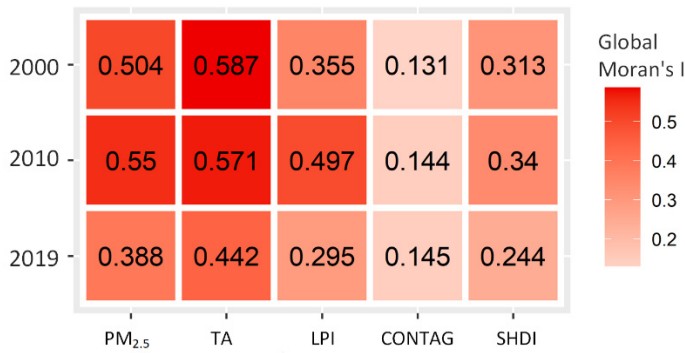

**Figure A1.** Global Moran's I of $PM_{2.5}$ and green space landscape indexes (all of the Moran's I values are significant at the level of 1%).

**Table A1.** Examples of $PM_{2.5}$ concentration guidelines.

| Organizations/ Commissions/ Countries | Guidelines for $PM_{2.5}$ ($\mu g/m^3$, Annual Average) | | Explanations | Reference |
|---|---|---|---|---|
| WHO | Interim target 1 | 35 | The Air quality guidelines (AQG) level was based on the relationship between PM2.5 and non-accidental mortality in the long run. The interim targets, as incremental steps, guide progressive air quality improvement for areas with high pollution. | [17] |
| | Interim target 2 | 25 | | |
| | Interim target 3 | 15 | | |
| | Interim target 4 | 10 | | |
| | AQG level | 5 | | |
| European Commissions | Target value to be met as of 1.1.2010; Limit value to be met as of 1.1.2015 | 25 | For a target value, countries are responsible to implement measures to ensure that it is attained. Limit value relates to the maximum margin of tolerance. Stage 2 is stricter than the former standards. | [20] |
| | Stage 2 limit value to be met as of 1.1.2020 | 20 | | |
| China | Level 1 | 15 | PM2.5 of natural reserve area and other protected places meet the Level 1; Residential area, factories, and rural area meet Level 2. No specific time limitation. | [18] |
| | Level 2 | 35 | | |
| India | National Ambient Air Quality Standard (NAAQS) | 40 | National standard released in 2009 | [19] |

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
