# Peer review of "Do We Need More Urban Green Space to Alleviate PM2.5 Pollution? A Case Study in Wuhan, China"

_land, doi:10.3390/land11060776_

Round 1
Reviewer 1 Report
Overall opinion:
The work presented here explores the possible relationship between urban green space and air pollution (considering in this case the species PM2.5). The main subject is the exploration of a land use database and annual means of PM2.5 concentrations from big data calculations, to find spatiotemporal connections.
The paper presents a very simplistic way of looking at the impact of urban green space, which is a very complex matter, but I understand the need to explore this kind of approach, especially when we speak about managing urban green areas scientists have to make the effort to convey simple tools to the stakeholders that very often are not scientists. Moreover it is my opinion that the largest the number of different approaches we use to study a problem the highest the probability of finally gaining a better understanding and hopefully solve it. I just don’t think that one can extract more information that what was put in and some of the conclusions of the paper are in my opinion at least a bit farfetched.
Therefore I think that the wording of some strong assessments made in the paper should be significantly re-dimensioned in order for the paper to be published. I also realize that the subject of the journal issue is not specifically Atmospheric Physics and that one cannot ask to add too much information about the complex atmospheric interaction that influence the air quality. Nevertheless it is important to remind the reader that there are multiple factors such as meteorology, sources, chemistry (partially already cited but I think they need to be stressed out more) and also biogenic emission (never mentioned in the paper).
Finally I list here my comments in the specific of some paragraphs:
Line 2 and 3. I think the title with a question is perfect for describing the work, which is exploring a correlation. This title is telling the viewer that the final goal is not necessarily to demonstrate that more urban green space means less PM2.5 but that we are studying to simplify something very complex.
Line 19 and 20. “Regression results show that large patches of green space significantly reduce PM2.5 concentration” this statement should be rephrased because if one does not identify (and therefore show) the physical mechanisms which relates the increase of the green space to the decrease of PM2.5 one cannot say that there is causality between these two. Since you only looked at land use and PM2.5 concentrations you can say that they are correlated, but you cannot say that one is the cause of the other. Moreover on May 2 2022 a paper was published on Environmental Science and Technology by Miao Yu et al, titled “Is urban greening an effective solution to enhance environmental comfort and improve air quality?” that is stating at page F: “Emissions from urban greening increased the concentration of PM2.5 remarkably. --- The effect of BVOC (Biogenic Volatile Organic Compounds) emissions due to urban greening on PM2.5 was obviously larger than that of land-use change”. I think this recent paper is of great interest to your work and should be cited and discussed in your paper too. It is based on the study of the atmosphere related to land use, employing an atmospheric transport model which takes into account the changes in land use and how they can effect meteorology and air quality in a urban area.
Lines 38 to 40. Measures aimed at improving air quality, such as reducing emissions, for example, must not be diminished because in my opinion they always remain the most effective and in some way the fastest for the reduction of pollutants in the atmosphere (banning vehicles circulation in cities can be faster than “growing” trees for example). From the point of view of urbanized management, the management of green spaces can help by having other positive implications such as ecosystem services. The question is how much can green spaces help air quality?
Lines 78 and 79. Here is another statement that can be controversial. Depending on the type of greening we can have different impact of particulate matter and this has been proven in many publications, as you state here. But there are also some mechanisms that you are completely neglecting in your research and you have to clearly state them (as for example the fact that plants emit chemical compounds that can influence the chemistry of the atmosphere and bring to an increase of the production of other pollutants).
Line 115. “a prior status” should be “a priority status”.
Line 120. “green space by land cover” should be “green space as composed by land cover…”.
Line 131. The description of the acronym GDP is missing.
Line 181. Here It’s the first time that SAR appears and should be written in its full form (Spatial Auto Regression) instead that in line 262.
Line 222. “dvelopment” should be “development”.
Line 269. What do the asterisks in the Table 4 indicate? I think it should be described in the caption of the table.
Line 276. I believe that the paper you refer as [54] studies the ability of plants to subtract PM2.5 but does not prove a decrease in PM2.5 concentration in cities.
Line 287. I could not find the value of correlation you cite here (0.293) in any of the tables and figures.
Line 290-291. Again this phrase is controversial and is not proven by your research. See my comment on Lines 38 to 40.
Line 295. My impression is that you did not cite properly paper [57]. The paper states that even if there is correlation between green spaces and PM2.5 there are some events in which this correlation is inverted, because the PM2.5 concentration can be influenced by other factors (as for example the variation of anthropogenic emissions). You should cite paper [57] as an example of the complexity you are trying to deal with.
Paragraph 4.3 It is good to have all the limitations of this approach listed in a paragraph and you should also add here the biogenic emissions and the complex chemistry happening in the atmosphere.
Line 336. You should substitute the phrase “there are few limitations of this work” with “there are some limitations to this work”.
Line 355. Substitute “migrate” with “decrease”.
Reviewer 2 Report
The review concerned an article entitled: Do we need more urban green space to alleviate PM2.5 pollution? A case study in Wuhan, China. Air pollution is still an important topic, especially in places with high pollution where phytoremediation as a solution could be used, so it is also important to verify and better select the greenery used in urban landscaping. In addition, the research concerns, on the PM2.5 fraction, which is of particular importance from the perspective of the population. Due to the fact that my native language is not English, I will not check the linguistic correctness of the manuscript. Overall the article is interesting and although it touches on an repeated topic it provides a refreshing approach to research on greenery and develop of the city on the air pollution.
Here are questions and suggestions:
Line 40-41. There is lack of citation of this information.
Line 44. I do not know why here and further the µg/m3 is written in italics.
Line 50 and 52 – Authors should write the decrease the life expectancy in months or days – It will be more representative and understandable
Line 100 – data like precipitation an temperature should be written to the first decimal place – 2012.4 and 17.2
Figure 1. – The Wuhan location in China should be add.
Line 198. Question for authors. What happened with this green spaces? Maybe explanation should be add to the manuscript.
Table 3. In the PM 2.5 – 2.5 should be in subscript
Line 294. Work abut different types of greenery in PM accumulation can be added here
Line 245, 259, 297, 299. In the PM 2.5 – 2.5 should be in subscript
Line 273 and 285. – Looking at China at this moment – with skyscrapers, fast rails, factories Etc. we cannot say that this country is still developing. Its fully developed. Comparing to other countries – eg in. Europe is developed time times more. The air pollution is also problem of developed countries.
Line 302-329 – This part has no discussion with other authors.
Line 349. I am suggesting year long intervals to understand the process.
Line 358, 259, 366, 299 in PM 2.5 – 2.5 should be in subscript
Reviewer 3 Report
Dear Authors,
Please consider the following comments as constructive contributions.
This paper needs extensive corrections of English language. Please consider the corrections identified in the attachment (It is not possible to reproduce these numerous language corrections needed here).
Furthermore, this paper presents some serious inconsistencies, which along with its very weak English, prevent publication. I will refer some of these aspects here.
line 51-52: Ref. 20 states that "If PM2.5 in all countries met the World Health Organization Air Quality Guideline (10 μg m–3), we estimate life expectancy could increase by a population-weighted median of 0.6 year." Thus, the claim in line 51-52 refers to the WHO guideline of 10 μg m–3 and not the most recent guideline value, 5 μg m–3 (refered in line 44).
Line 78: It should not be refered as a "negative impact of green space on PM2.5", since it is misleading.
Line 84-85: The sentence does not make sense.
Line 92-93: At this point of the paper, it is not clear what is meant by these "cold spots of green space landscape in the process of alleviating PM2.5" and thus clarification is mandatory.
Line 93: The claim of having "proposed strategies to enhance the relashion between PM2.5 and green space" is not completely fulfilled in Discussion and Conclusions and should be further developed.
Line 121: A reference must be included
Line 127-129: The time scale of the PM2.5 concentration data extracted from database must be specified (daily, monthly or yearly 1 km average)
Line 129: Ref. 46 is not correct.
Line 134-137: This paragraph, which refers the tools that have been used (ArcGIS, Stata15 etc), should be in 2.3. Methodology.
Line 140-142: References 47 and 48 do not make sense here, since these aspects should be addressed in the Introduction. Here in the Methodology, references should be specific for the methodological tools used.
Table 1: At least one reference must be included for each Index refered in Table 1.
Table 1: CONTAG Index expression in not correct.
Table 1: The explanation of the meaning of the CONTAG index is not clear. I suggest: "The lower the CONTAG index value, the more scattered the urban landscape pattern and the higher the average degree of fragmentation."
Line 152: Ref. 49 must be incorrect.
Line 153: Moran's I appears incorrectly refered as "Moran'I" in all the document.
Line 154: A reference should be included.
Line 162: A reference should be included.
Line 166: The adopted confidence level should be refered.
Line 183: A reference should be included and it should be specified what "SAR" stands for. The definition of rho should be presented here, where it is first mentioned.
Line 186: A reference should be included.
Line 189: Spatial Durbin Model appears incorrectly refered to as "Spatial Durban Model" (typo).
Line 196 and subchapter 3.1.: Since for PM2.5, data of 2019 is used instead of 2020, the same criterion should be adopted in what concerns green space spatial change (Figure 2.) and the calculation of lansdscape indexes (Table 2), for coherence.
Line 212: Figure captions must give complete information, including location and source of the data (when applicable).
Table 2: Remove "100" in "LPI (100%)", "CONTAG (100%)" and "changing rate (100%)"
Line 215: From Figure 3, is an overall decrease of PM2.5 concentrations from 2000 to 2010 is apparent, but this is contradicted by the boxplot in Figure A2 and must be addressed.
Line 216-219: While aware that the air quality situation in China is different from other parts of the world, it is incomprehensible that PM2.5 concentrations of 36-75 µg.m-3 are refered to as "fine air quality" and concentrations 76-115 µg.m-3 as "slight pollution", when these concentrations are very high. For instants, in EU, Directive 2008/50/EC determines a PM2.5 annual limit of 20 µg.m-3 and the WHO annual guideline value is 5 µg.m-3.
The threshold values of the Chinese air quality standards must be correctly specified in the Introduction and the qualification of the concentrations values refered in lines 216, 218 and 219 must be done in exact agreement with the standard.
Furthermore, these values should be put in perspective, refering that they are much higher than other international standards (e.g. EU Directive 2008/50/EC and WHO guidelines).
Line 218: It is stated that "The concentration of PM2.5 significantly decreased from 2010 to 2019". Statistical test has been done? Which test? (specify in Methodology). If not, remove "significantly".
Line 227: Figure captions must give complete information, including location and source of the data.
Figure 3.: Legends must include indication of the represented variable (PM2.5) and units. The intervals of concentrations represented by each color must be specified.
Figure 3. Strangely the legend refers negative PM2.5 concentration values (in blue)...
Line 241-244: The increase in the low-low groups in the left side of the Yangtze River should be better explained, since the stated economic development would in principle lead to higher PM2.5 concentrations, in the presence of lower Total Areas of green spaces.
Figure 4. The scale in Figure 4 Legend, is wrong...
Line 251: Abbreviations (VIF, LR, AIC, BIC) should be explained (e.g. in Methodology)
Line 253-254: Insert references.
Line 258: It is stated that artificial surface can explain PM2.5 concentrations, but artificial surface appears with coefficient 0.000 in Table 4...
Table 4: The significant negative relation between GDP and PM2.5 should be commented and should be contextualized in the light of the literature. Would it be expected? How can it be explained?
Line 260: The influence of SHDI must also be tentatively explained.
Line 263: It is stated that TA significantly correlate with PM2.5, but in Table 4 TA does not present * in SAR (??)
Line 269 (Table 4 caption): Regression analysis of what in terms of what (must be specified). Significance of ** and *** must be refered in the caption. The time span of the regression analysis must also be specified (according to line 289, it seems it is 2010-2020...)
Chapter 3.4.: There is no comment or discussion on SEM and SDM results and these, once included in Table 4, should be analysed.
Line 271: It is stated that "Shrinking green space poses challenges in alleviating PM2.5 concentration", but according to SAR, SEM and SDM, this conclusion seems not valid, since the TA coefficients are positive...
Line 272-279: Line 272 to 279 does not make sence here, since these are generic aspects already addressed in the Introduction.
Line 283-286: The meaning of the sentences is not understandable. Rephrase or remove.
Line 303: This sentence is not understandable.
Line 308-310: This sentence is not understandable.
Line 315: It is stated that "The most apparent alternation (??) appeared in the southwestern part of the central city. New built residential areas expanded the impervious surface scale while invading green space." I can not identify this trend in Figure 4...
Line 319-320: In the northern part of the city, what is seen in Figure 4 is the appearance of High-High spots (high PM2.5 with high TA of green space, and not "high PM2.5 concentration with less green space" as stated).
Line 337: The reasons for considering PM2.5 data of 2019 instead of 2020 must be explained much earlier in the paper, namely in 2.2. Data for analysis.
Line 360: Only PM2.5 is addressed in this paper (so "air pollutants" must be substituted by "PM2.5 concentrations").
Line 367: The paper does not present results relative to biodiversity, so this conclusion must be rephrased.
Figure A1: I do not understand what is represented in these graphs, since the values of the Indexes do not correspond to what is refered in Table 1...
Please adress these inconsistencies.
Axis labels and units (when applicable) must also be specified .
Line 373 (Figure A1 caption): The constitution of the boxplot must be explained in the caption (median and quartiles represented by the wiskers and the box, outliers)
Figure A2: Insert axis label (with units)
PM2.5 data for 2019 must be represent instead of 2020, given the option expressed by the authors in line 337.
References: Several DOIs or hiperlinks are incorrect
Reference 49 is repeated (ref 40).
In order to be able to be published, the refered aspects (and all the aspects refered in the attached file) must be fully addressed and the paper must be subjected to significant modifications.
Best regards.

Author Response
Response to reviewer 3 comments
Comment 1
Dear Authors,
Please consider the following comments as constructive contributions.
This paper needs extensive corrections of English language. Please consider the corrections identified in the attachment (It is not possible to reproduce these numerous language corrections needed here).
Furthermore, this paper presents some serious inconsistencies, which along with its very weak English, prevent publication. I will refer some of these aspects here.
line 51-52: Ref. 20 states that "If PM2.5 in all countries met the World Health Organization Air Quality Guideline (10 μg m–3), we estimate life expectancy could increase by a population-weighted median of 0.6 year." Thus, the claim in line 51-52 refers to the WHO guideline of 10 μg m–3 and not the most recent guideline value, 5 μg m–3 (refered in line 44).
Response 1
Thanks for pointing it out.
(1) We added the explanation that the air quality guideline was released in 2005 (Line 53).
(2) The differences of air quality guideline were shown in Appendix Table A1 (Line 403).
Comment 2
Line 78: It should not be refered as a "negative impact of green space on PM2.5", since it is misleading.
Response 2
Revised (Lines 72-74).
We replaced the controversial expression with the way of green space influences PM2.5 concentration.
Comment 3
Line 84-85: The sentence does not make sense.
Response 3
Revied and removed.
Comment 4
Line 92-93: At this point of the paper, it is not clear what is meant by these "cold spots of green space landscape in the process of alleviating PM2.5" and thus clarification is mandatory.
Response 4
Revised (Lines 100-101).
The main object of this manuscript is the correlation between green space landscape patterns and PM2.5 concentration. Accordingly, we revised the expression here.
Comment 5
Line 93: The claim of having "proposed strategies to enhance the relashion between PM2.5 and green space" is not completely fulfilled in Discussion and Conclusions and should be further developed.
Response 5
Thanks for pointing it out.
We added the possible strategies to preserve and enhance large patches of green space in discussion (Lines 363-365) and conclusion (Line 397-398).
Comment 6
Line 121: A reference must be included
Response 6
Added reference [54] (Line 128).
Comment 7
Line 127-129: The time scale of the PM2.5 concentration data extracted from database must be specified (daily, monthly or yearly 1 km average)
Response 7
Added (Line 134).
PM2.5 concentration data with 1km resolution is annual average.
Comment 8
Line 129: Ref. 46 is not correct.
Response 8
Revised and we checked all of the references.
Comment 9
Line 134-137: This paragraph, which refers the tools that have been used (ArcGIS, Stata15 etc), should be in 2.3. Methodology.
Response 9
Added (Lines 155-157; Lines 226-227).
Comment 10 and 11
Line 140-142: References 47 and 48 do not make sense here, since these aspects should be addressed in the Introduction. Here in the Methodology, references should be specific for the methodological tools used.
Table 1: At least one reference must be included for each Index refered in Table 1.
Response 10 and 11
Revised (Line 148-154).
(1) We removed the sentences and added references for each landscape index;
(2) Table 1 was revised.
Comment 12
Table 1: CONTAG Index expression in not correct.
Table 1: The explanation of the meaning of the CONTAG index is not clear. I suggest: "The lower the CONTAG index value, the more scattered the urban landscape pattern and the higher the average degree of fragmentation."
Response 12
Revised (Table 1 158).
Comment 13
Line 152: Ref. 49 must be incorrect.
Response 13
Thanks for the comment. We have removed the reference.
Comment 14
Line 153: Moran's I appears incorrectly referred as "Moran'I" in all the document.
Response 14
We apologize for the typo. We revised it in the manuscript.
Comment 15 and 16
Line 154: A reference should be included.
Line 162: A reference should be included.
Response 15 and 16
Added (Lines 172 and 186).
Comment 17
Line 166: The adopted confidence level should be refered.
Response 17
Added (Line 191). The confidence level in this study is 5%.
Comment 18
Line 183: A reference should be included and it should be specified what "SAR" stands for. The definition of rho should be presented here, where it is first mentioned.
Line 186: A reference should be included.
Response 18
Revised (Lines 206; 210).
(1) According to previous researches (Golgher and Voss, 2016; Lee and Yu, 2010) , spatial lag model (SAR) aims to manifest the effect of lagged dependent variables. We explained the meaning and chose “spatial lag model (SLM)” for the manuscript.
(2) We added the definition of rho at the first time it appeared.
(3) Two references were added (Golgher and Voss, 2016; Lee and Yu, 2010).
Comment 19
Line 189: Spatial Durbin Model appears incorrectly refered to as "Spatial Durban Model" (typo).
Response 19
Revised (Line 215).
Comment 20
Line 196 and subchapter 3.1.: Since for PM2.5, data of 2019 is used instead of 2020, the same criterion should be adopted in what concerns green space spatial change (Figure 2.) and the calculation of lansdscape indexes (Table 2), for coherence.
Response 20
Thanks for the suggestions. We explained the reason of adopting the year of green space spatial change (Lines 132-133), and revised the graphs and tables in the manuscript.
Comment 21
Line 212: Figure captions must give complete information, including location and source of the data (when applicable).
Response 21
Added (Lines 250-252).
Comment 22
Table 2: Remove "100" in "LPI (100%)", "CONTAG (100%)" and "changing rate (100%)"
Response 22
Revised Table 2 (Line 246).
Comment 23
Line 215: From Figure 3, is an overall decrease of PM2.5 concentrations from 2000 to 2010 is apparent, but this is contradicted by the boxplot in Figure A2 and must be addressed.
Response 23
Revised (Figure 3, Line 264).
(1) Figure 3 was generated from ArcGIS (10.5), layer properties’ “Symbology”. “Standard Deviation” was the stretch type, but the maximum and minimum values of PM2.5 concentration were different in 2000, 2010, and 2019. So, we revised the spatial distribution of PM2.5 by changing the stretch type into “Minimum Maximum”. It Applies a linear stretch based on the output minimum and output maximum pixel values, which are used as the endpoints for the histogram (ArcGIS Pro help, https://pro.arcgis.com/en/pro-app/latest/help/data/imagery/symbology-pane.htm). The minimum and maximum values are 0 and 140.
(2) Appendix Figure A2 was combined with Figure 3 to form a new graph.
Comment 24
Line 216-219: While aware that the air quality situation in China is different from other parts of the world, it is incomprehensible that PM2.5 concentrations of 36-75 µg.m-3 are refered to as "fine air quality" and concentrations 76-115 µg.m-3 as "slight pollution", when these concentrations are very high. For instants, in EU, Directive 2008/50/EC determines a PM2.5 annual limit of 20 µg.m-3 and the WHO annual guideline value is 5 µg.m-3.
The threshold values of the Chinese air quality standards must be correctly specified in the Introduction and the qualification of the concentrations values referred in lines 216, 218 and 219 must be done in exact agreement with the standard.
Furthermore, these values should be put in perspective, referring that they are much higher than other international standards (e.g. EU Directive 2008/50/EC and WHO guidelines).
Response 24
Thanks for putting forward the important point.
(1) We added a list of air quality guidelines of PM2.5 concentration (Appendix Table A1, Line 403).
(2) The grade of PM2.5 concentration in lines 216, 218, and 219 was removed (Line 254).
Comment 25
Line 218: It is stated that "The concentration of PM2.5 significantly decreased from 2010 to 2019". Statistical test has been done? Which test? (specify in Methodology). If not, remove "significantly".
Response 25
Revised (Line 257).
Comment 26
Line 227: Figure captions must give complete information, including location and source of the data.
Response 26
Added (Line 264).
Comment 27
Figure 3.: Legends must include indication of the represented variable (PM2.5) and units. The intervals of concentrations represented by each color must be specified.
Response 27
Revised (Figure 3, Line 264).
(1) We added “PM2.5 concentration” and units in Figure 3.
(2) We tried to show the intervals of PM2.5 concentrations by applying “Graduated colors” in Symbology. It seems hard to distinguish the differences of PM2.5 for each grid. So, we changed the stretch type into “Minimum-Maximum”. As the response 23 has mentioned, “Minimum Maximum” stretch type might be better to reflect the differences of PM2.5 concentrations in the central city area. We combined the boxplot (Originally it was Appendix Figure A1) to demonstrate the spatial trend of PM2.5.
Comment 28
Figure 3. Strangely the legend refers negative PM2.5 concentration values (in blue)...
Response 28
Thanks for the question. To make the spatial trend of PM2.5 concentration more clear, we replaced the “2010-2019” PM2.5 concentration into the boxplot (Figure 3, Line 264).
Comment 29
Line 241-244: The increase in the low-low groups in the left side of the Yangtze River should be better explained, since the stated economic development would in principle lead to higher PM2.5 concentrations, in the presence of lower Total Areas of green spaces.
Response 29
Revised in discussion (Line 331-339).
According to the Environmental Kuznets Curve, the post urbanization period economic development can exert a less negative influence on the environment. In our study, during the primary period of urbanization, economic development dominated society. Factories with polluting emissions near the central city center led to poor air quality. Later, advanced economic development and urban planning will reduce the disturbance to air quality. This interpretation might explain the cluster of the “L-L” group on the left side of the Yangtze River where high urbanization and low area of green space occur.
Comment 30
Figure 4. The scale in Figure 4 Legend, is wrong...
Response 30
Revised (Line 280).
Comment 31
Line 251: Abbreviations (VIF, LR, AIC, BIC) should be explained (e.g. in Methodology)
Response 31
Added in Methodology (Lines 221-225).
Comment 32
Line 253-254: Insert references.
Response 32
Added (Lines 222-225).
Comment 33
Line 258: It is stated that artificial surface can explain PM2.5 concentrations, but artificial surface appears with coefficient 0.000 in Table 4...
Response 33
We apologize for the mistake. We revised the expression into “Artificial surface demonstrates an insignificant correlation with PM2.5” (Line 292).
Comment 34
Table 4: The significant negative relation between GDP and PM2.5 should be commented and should be contextualized in the light of the literature. Would it be expected? How can it be explained?
Response 34
Revised in discussion (Line 331-339).
Comment 35
Line 260: The influence of SHDI must also be tentatively explained.
Response 35
Added (Line 355-356).
Comment 36
Line 263: It is stated that TA significantly correlate with PM2.5, but in Table 4 TA does not present * in SAR (??)
Response 36
We apologize for the mistake. We revised the statement (Line 305).
Comment 37
Line 269 (Table 4 caption): Regression analysis of what in terms of what (must be specified). Significance of ** and *** must be refered in the caption. The time span of the regression analysis must also be specified (according to line 289, it seems it is 2010-2020...)
Chapter 3.4.: There is no comment or discussion on SEM and SDM results and these, once included in Table 4, should be analyzed.
Response 37
Thanks for the suggestions.
(1) We identified the independent variables and dependent variables, and added the time span in the caption of Table 3 (Table 4 in the original manuscript), Line 307.
(2) We added a note below Table 3 to explain the significant level of asterisks (Line 309).
(3) We added the analysis of SEM and SDM results (Line 298-300).
Comment 38
Line 271: It is stated that "Shrinking green space poses challenges in alleviating PM2.5 concentration", but according to SAR, SEM and SDM, this conclusion seems not valid, since the TA coefficients are positive...
Response 38
Revised (Lines 311-315).
We reorganized the discussion. The correlation of large green space patches and PM2.5 concentration was put in the first section.
Comment 39
Line 272-279: Line 272 to 279 does not make sence here, since these are generic aspects already addressed in the Introduction.
Response 39
Revised (Lines 311-385).
We revised the discussion according to the results. The discussion mainly includes the relationship between large green space patches and PM2.5, the negative correlation of GDP and PM2.5 concentration, and limitations.
Comment 40
Line 283-286: The meaning of the sentences is not understandable. Rephrase or remove.
Line 303: This sentence is not understandable.
Line 308-310: This sentence is not understandable.
Response 40
Thanks for the comments.
We removed the these sentences and restructured the discussion (Line 311-330).
Comment 41
Line 315: It is stated that "The most apparent alternation (??) appeared in the southwestern part of the central city. New built residential areas expanded the impervious surface scale while invading green space." I can not identify this trend in Figure 4...
Response 41
We agree with the reviewer’s comment. The sentence has been removed.
Comment 42
Line 319-320: In the northern part of the city, what is seen in Figure 4 is the appearance of High-High spots (high PM2.5 with high TA of green space, and not "high PM2.5 concentration with less green space" as stated).
Response 42
Revised (337-339).
Comment 43
Line 337: The reasons for considering PM2.5 data of 2019 instead of 2020 must be explained much earlier in the paper, namely in 2.2. Data for analysis.
Response 43
Revised (Line 131-132).
Comment 44
Line 360: Only PM2.5 is addressed in this paper (so "air pollutants" must be substituted by "PM2.5 concentrations").
Response 44
Revised (Line 387).
Comment 45
Line 367: The paper does not present results relative to biodiversity, so this conclusion must be rephrased.
Response 45
Thanks for the suggestions. We have removed the content.
Comment 46
Figure A1: I do not understand what is represented in these graphs, since the values of the Indexes do not correspond to what is refered in Table 1...
Please adress these inconsistencies.
Axis labels and units (when applicable) must also be specified .
Response 46
Revised.
Figure A1 was different with Table 1 because of the scale. Figure A1 was based on the grid scale, while Table 1 was based on the research area. We tried to comprehensively show the temporal changes of green space landscape patterns from two different scales. But it seems to provide confusing information, we decided to remove the Figure A1.
Comment 47
Line 373 (Figure A1 caption): The constitution of the boxplot must be explained in the caption (median and quartiles represented by the wiskers and the box, outliers)
Response 47
Revised.
The Figure A1 has been removed for the clear expression.
Comment 48
Figure A2: Insert axis label (with units)
PM2.5 data for 2019 must be represent instead of 2020, given the option expressed by the authors in line 337.
Response 48
Revised (Figure 3, Line 264).
(1) We combined Figure A2 with Figure 3 to show the spatial and temporal changes of PM2.5 concentration. The axis label with units have been added.
(2) The time of PM2.5 data has been changed into 2019.
Comment 49
References: Several DOIs or hiperlinks are incorrect
Reference 49 is repeated (ref 40).
Response 49
Revised and checked the reference in the manuscript.
Comment 50
In order to be able to be published, the refered aspects (and all the aspects refered in the attached file) must be fully addressed and the paper must be subjected to significant modifications.
Best regards.
Response 50
We deeply appreciate your insightful comments and suggestions. After the major revision, the manuscript was polished by the native English speaker.
Reference
[1] Golgher, A.B.; Voss, P.R. How to Interpret the Coefficients of Spatial Models: Spillovers, Direct and Indirect Effects. Spat. Demogr. 2016, 4, 175–205. https://doi.org/10.1007/s40980-015-0016-y.
[2] Lee, L.; Yu, J. Estimation of spatial autoregressive panel data models with fixed effects. J. Econom. 2010, 154, 165–185. https://doi.org/https://doi.org/10.1016/j.jeconom.2009.08.001.
We sincerely thank you for the constructive comments and suggestions. We carefully read the PDF. Those suggestions and comments that didn’t list above are also updated in the manuscript.
Best regards,
All of the authors
Reviewer 4 Report
Review Ref.: land-1718849
The manuscript titled “Do we need more urban green space to alleviate PM2.5 pollution? A case study in Wuhan, China” analyzes spatiotemporal trends of green areas by different landscape indexes and their relationships with PM2.5 levels in latest twenty years in the rapid growing city of Wuhan (China).
The topic is interesting and the manuscript is well-structured and written in all parts. The research question is properly outlined and exhaustively addressed.
I think it is a good work, deeply and comprehensively discussed in various aspects. However, some little revisions are needed before being accepted for publication. Therefore, in my opinion, it could be considered after some adjustments and corrections as indicated in the following.
Major comments
- The background knowledge about the topic is explained in the introduction. However, a deeper review in the discussion section, also with quantitative estimates of air pollution improvements related to green space-based policies could be useful to make results analysis more robust and comparable with other outcomes. Please, add this aspect.
- What vegetation types are considered as “green space”?
- It could be better to add a land use map as figure, representing, for example, land use change rate (%) between different decades. In this way, information about correlation between PM2.5 level and vegetation type could be evaluated. In this way, focused plant species could be suggested in order to obtain maximum efficiency in lowering particulate concentrations in differently polluted areas.
- Could such analysis be performed with other air pollutants?
Minor corrections
- Line 36: why is PM2.5 defined as “unexpected product of industrialization…”?
- Line 122: modify “21010” with “2010”
- Line 136: modify with “girds” with “grids”
- Line 141: delete “pp. 330-351”
- Line 171: what’s about “queen adjacency”? is it a mistake?
Reviewer 5 Report
General comment
The paper reports an analysis of the effect of green space on PM2.5 concentrations in Wuhan (China) discussing to a certain extent the trends between 2010 and 2019. The topic is interesting and suitable for the Journal. However, methodology has several limitations that are not clearly discussed and confounding factors are not taken into account. This limit the usefulness of the conclusions (see my specific comments). In addition, other aspects are not completely clear. I suggest to consider the paper for publication only after a major revision that addresses my specific comments.
Specific comments
The paper puts in correlation some statistical properties of green space with PM2.5 concentration trends in time and space. However, the trends of concentration are strongly influenced by changes of source as well as location (i.e. distance from sources) and so on. It seems that these confounding factors are not taken into account in the analysis and all changes are considered and changes are only considered explainable by green space. This could lead to erroneous conclusions. Please explain why such limited number of parameters limited to green spaces are used in the analysis.
Related to the previous point. It appears that there is a continuous decrease of green space but also a relevant decrease of PM2.5. This seems to contradict your conclusion that green spaces are relevant to mitigate PM2.5 pollution. This aspect should be explained in detail. Furthermore, from 2010 to 2019 the substantial drop of PM2.5 that is likely due to mitigation strategies and policies on emissions rather than on green spaces. This should be clearly mention and explained in discussion and in the conclusions. It could also be suggested that these strategies should be pursued further to reduce atmospheric pollution and that green spaces could give a minor contribution compared to those policies.
Line 16. Better maximum rather than summit
Line 36. I do not believe that this is unexpected considering that it is known since several years that anthropogenic sources have relevant effects on PM2.5 levels.
Introduction. It could be mentioned that measurements of particle fluxes in urban environments put in evidence an increase of deposition in vegetated areas, see for example Contini et al. (Atmospheric Environment 46, 2012, 25-35) and Casquero-Vera (Atmospheric Environment 269, 2022, 118849).
Figure A2 indicates 2020 in the rest of the paper it is used 2019, likely for the effect of COVID-19 lockdown and restriction measures. Please correct this incongruence.
Table 3. The column of p-value with all zero is meaningless. Please use the correct notation or the correct number of decimal digits to provide an useful information.
Figure 4 and related text. It is not clear what is low-low, low-high and so on. It is referred to certain thresholds of correlation coefficients? What does the numbers reported in the small legends of the graphs?
Lines 324-326. It seems strange that high energy consumption is promoted for sustainable development. Please check this sentence.
Lines 342-346. It is true that after lockdown concentrations raised, however, the effects of COVID-19 policies had in several places positive effects also after the lockdown period at least for several months, see for example, the analysis of Dinoi et al. (Atmosphere 2021, 12, 352). This aspect should be mentioned.
Figure A1. It is not clear what are the units of the vertical axis. I would expect, for example, TA as a surface m2 for example. Please add units.
Line 70. Please remove etc. It is better to mention explicitly the different aspects that authors consider relevant.
There are several instances in which PM2.5 is written without subscript. Please correct this aspect.
Table 1. Second row. What is A? It is TA?
Line 136. Grids rather than girds.
Line 146. Please use apex for m2.
Line 354. Better to say deposition on leaves.
Line 355. Mitigate rather than migrate.
Author Response
Response to reviewer 5 comments
General comment
The paper reports an analysis of the effect of green space on PM2.5 concentrations in Wuhan (China) discussing to a certain extent the trends between 2010 and 2019. The topic is interesting and suitable for the Journal. However, methodology has several limitations that are not clearly discussed and confounding factors are not taken into account. This limit the usefulness of the conclusions (see my specific comments). In addition, other aspects are not completely clear. I suggest to consider the paper for publication only after a major revision that addresses my specific comments.
Comment 1
Specific comments
The paper puts in correlation some statistical properties of green space with PM2.5 concentration trends in time and space. However, the trends of concentration are strongly influenced by changes of source as well as location (i.e. distance from sources) and so on. It seems that these confounding factors are not taken into account in the analysis and all changes are considered and changes are only considered explainable by green space. This could lead to erroneous conclusions. Please explain why such limited number of parameters limited to green spaces are used in the analysis.
Response 1
The R-square of OLS is 0.435 in the present study. There are other variables not included, but it can examine the correlation between green space morphology and PM2.5 concentration under certain preconditions.
Air purification of green space could directly remove the particulate matter. However, the amount of green space was in downward trend in the central city of Wuhan. Increment and protection of green space will improve the air purification and other ecosystem services. Urban green space landscape patterns are formed in the process of increment and protection. So, we analyzed the better landscape indexes from the perspective of correlating with PM2.5 concentration.
Except for green space factors, we also added GDP as a influential variable to reflect the economic intensity. It would be better to take the location of sources into consideration, which we didn’t because of the data availability. We might search for the data and expand it in the following work.
Comment 2
Related to the previous point. It appears that there is a continuous decrease of green space but also a relevant decrease of PM2.5. This seems to contradict your conclusion that green spaces are relevant to mitigate PM2.5 pollution. This aspect should be explained in detail. Furthermore, from 2010 to 2019 the substantial drop of PM2.5 that is likely due to mitigation strategies and policies on emissions rather than on green spaces. This should be clearly mention and explained in discussion and in the conclusions. It could also be suggested that these strategies should be pursued further to reduce atmospheric pollution and that green spaces could give a minor contribution compared to those policies.
Response 2
Revised (Lines 320-327; 331-340).
We revised the large green space patches as the supplementary measures for PM2.5 mitigation.
Comment 3
Line 36. I do not believe that this is unexpected considering that it is known since several years that anthropogenic sources have relevant effects on PM2.5 levels.
Response 3
Revised. The expression has been revised as “Small particulate matter with widths less than 2.5 micrometers (PM2.5) is one of the outcomes of anthropogenic activities …” (Line 35).
Comment 4
Introduction. It could be mentioned that measurements of particle fluxes in urban environments put in evidence an increase of deposition in vegetated areas, see for example Contini et al. (Atmospheric Environment 46, 2012, 25-35) and Casquero-Vera (Atmospheric Environment 269, 2022, 118849).
Response 4
Thanks for your suggestions.
We added the paper focusing on particle fluxes in urban environments (Line 350).
Comment 5
Figure A2 indicates 2020 in the rest of the paper it is used 2019, likely for the effect of COVID-19 lockdown and restriction measures. Please correct this incongruence.
Response 5
Revised.
To demonstrate the spatial and temporal changes of PM2.5 concentration clearer, we have combined Figure A2 with Figure 3 (Line 264).
Comment 6
Table 3. The column of p-value with all zero is meaningless. Please use the correct notation or the correct number of decimal digits to provide an useful information.
Response 6
Thanks for the suggestions. We transformed Table 3 into Figures (Appendix Figure A1, Line 401)
Comment 7
Figure 4 and related text. It is not clear what is low-low, low-high and so on. It is referred to certain thresholds of correlation coefficients? What does the numbers reported in the small legends of the graphs?
Response 7
Revised.
(1) We added the description of low-low and low-high in methodology (Lines 191-195).
(2) The correlation coefficients are significant at 5%. We added explanations on the caption of Figure 4 (Line 280-283).
Comment 8
Lines 324-326. It seems strange that high energy consumption is promoted for sustainable development. Please check this sentence.
Response 8
Revised (Line 325-327).
Comment 9
Lines 342-346. It is true that after lockdown concentrations raised, however, the effects of COVID-19 policies had in several places positive effects also after the lockdown period at least for several months, see for example, the analysis of Dinoi et al. (Atmosphere 2021, 12, 352). This aspect should be mentioned.
Response 9
Revised and cited (Lines 373).
Comment 10
Figure A1. It is not clear what are the units of the vertical axis. I would expect, for example, TA as a surface m2 for example. Please add units.
Response 10
Figure A2 was the other format of Table 2. We thought it might be better to remove the Figure A2 in the present version to avoid redundancy.
Comment 11
Line 70. Please remove etc. It is better to mention explicitly the different aspects that authors consider relevant.
Response 11
Removed (Lines 73-74).
Comment 12
Table 1. Second row. What is A? It is TA?
Response 12
Added (Line 160).
Comment 13
Line 16. Better maximum rather than summit
Response 13
Revised (Line 17).
Comment 14
Line 136. Grids rather than girds.
Response 14
Revised (Line 156).
Comment 15
Line 146. Please use apex for m2.
Response 15
Revised (Line 160).
Comment 16
Line 354. Better to say deposition on leaves.
Response 16
Revised (Line 383).
Comment 17
Line 355. Mitigate rather than migrate.
Response 17
Revised (Line 384).
Comment 18
There are several instances in which PM2.5 is written without subscript. Please correct this aspect.
Response 18
Revised and checked in the manuscript.
Thanks for your insightful comments and helpful suggestions.
Best regards,
All the authors.
Round 2
Reviewer 3 Report
Dear Sirs
The paper has been significantly improved and deserves publication, but still needs an English language revision (e.g. line 323 "Anthropogenic emission abatements dominants the contributions.").
Best regards
Author Response
Point 1: The paper has been significantly improved and deserves publication, but still needs an English language revision (e.g. line 323 "Anthropogenic emission abatements dominants the contributions.").
Response 1: All authors sincerely thank you for your suggestion. We checked expressions in the manuscript and asked the proofreading service to polish the language.
Best regards,
All the authors
Reviewer 5 Report
Authors improved the paper during revision and addressed my questions. I suggest to accept it for publication in the current form.
Author Response
Thanks for your comment.
Best regards,
All the authors